# Mucormycosis in CAPA, a Possible Fungal Super-Infection

**DOI:** 10.3390/jof7090708

**Published:** 2021-08-28

**Authors:** Paola Saltini, Emanuele Palomba, Valeria Castelli, Marco Fava, Laura Alagna, Simona Biscarini, Marco Mantero, Francesco Blasi, Anna Grancini, Alessandra Bandera, Andrea Gori, Antonio Muscatello, Andrea Lombardi

**Affiliations:** 1Infectious Diseases Unit, Fondazione IRCCS Ca’ Granda Ospedale Maggiore Policlinico, 20122 Milano, Italy; paola.saltini@unimi.it (P.S.); emanuele.palomba@unimi.it (E.P.); valeria.castelli@unimi.it (V.C.); marco.fava@unimi.it (M.F.); laura.alagna@policlinico.mi.it (L.A.); simona.biscarini@policlinico.mi.it (S.B.); alessandra.bandera@unimi.it (A.B.); andrea.gori@unimi.it (A.G.); antonio.muscatello@policlinico.mi.it (A.M.); 2Department of Pathophysiology and Transplantation, University of Milano, 20122 Milano, Italy; marco.mantero@unimi.it (M.M.); francesco.blasi@unimi.it (F.B.); 3Centre for Multidisciplinary Research in Health Science (MACH), University of Milano, 20122 Milano, Italy; 4Respiratory Unit and Cystic Fibrosis Adult Center, IRCCS Fondazione IRCCS Ca’ Granda Ospedale Maggiore Policlinico, 20122 Milano, Italy; 5UOS Microbiology, Central Laboratory, IRCCS Foundation, Cà Granda Ospedale Maggiore Policlinico, 20122 Milano, Italy; anna.grancini@policlinico.mi.it

**Keywords:** mucormycosis, CAPA, COVID-19

## Abstract

The occurrence of pulmonary fungal superinfection due to *Aspergillus* spp. in patients with COVID-19 is a well-described complication associated with significant morbidity and mortality. This can be related to a directed effect of the virus and to the immunosuppressive role of the therapies administered for the disease. Here, we describe the first case of pulmonary infection due to Mucorales occurring in a patient with a concomitant diagnosis of COVID-19-associated pulmonary aspergillosis.

## 1. Introduction

Since December 2019, the coronavirus disease 2019 (COVID-19) quickly spread worldwide and was declared as a pandemic by the World Health Organization (WHO) on 11 March 2020 [1]. The disease pattern of COVID-19 can range from mild to life-threatening pneumonia, with worst outcomes in elder patients and people with comorbidities such as severe asthma, diabetes mellitus or cancer [2]. The direct damage of the airway epithelium caused by the virus, the severe lymphopenia and the immune dysfunction induced by the infection [3,4] facilitate bacterial and fungal superinfection. Moreover, patients requiring oxygen or ventilatory support are currently managed with systemic glucocorticoids, the only treatment which has been demonstrated to reduce mortality [5,6], but steroid administration is a well-known risk factor for infections [7]. Usually, steroid therapy is administered as dexamethasone 6 mg/day based on the results of the RECOVERY trial [8], but other immunomodulatory molecules with immunosuppressive action are widely used, even if the efficacy of methylprednisolone and hydrocortisone has not been so clearly demonstrated [9,10,11]. Opportunistic fungal infections are of concern in such patients as an additional contributing factor to mortality.

There have been several reports of COVID-19-associated pulmonary aspergillosis (CAPA) [12] and some cases of mucormycosis, especially in patients with diabetes [13,14]. The diagnosis of COVID-19-associated mucormycosis (CAM) is more challenging because it is less frequently suspected by the clinicians and the causative agents are more difficult to isolate in cultures. Furthermore, the diagnosis cannot be helped by biomarkers such as β-D-glucan and galactomannan, as it is possible for aspergillosis [15,16].

Here we report the first case of the coexistence of two destructive fungal diseases, pulmonary aspergillosis and mucormycosis, in a patient with COVID-19 who was treated with corticosteroids. Our case emphasizes the importance of the clinical suspicion of invasive mould disease in this setting and the need to repeat microbiological investigations on bronchoalveolar lavage or other respiratory specimens, especially in case of poor response to treatment.

## 2. Materials and Methods

In our laboratory, respiratory samples of patients at risk for fungal infections, such as patients hospitalized in the intensive care unit, are routinely investigated for the presence of fungi. Our protocols include the culture of respiratory samples on Sabouraud Dextrose Agar, incubated at 32 °C for 5–20 days.

### Genus and Species Identification and Susceptibility Testing

Identification to genus and species level is strongly recommended for improved epidemiological data and to guide the choice of the antifungal treatment. We employed matrix assisted laser desorption ionisation time of flight (MALDI-TOF; Vitek MS BioMèrieux–IVD; Liyon, France) together with the macro and microscopic morphological evaluation of the fungus. The commercial methods E-test (BioMèrieux) with RPMI-1640 medium (BioMèrieux) was used for the antifungal susceptibility. For Aspergillus spp., the EUCAST clinical breakpoint for fungi was employed [17]. No interpretative clinical breakpoints are available for Mucorales spp., and the classification of isolates as susceptible or resistant is not possible.

## 3. Case Presentation

A 72-year-old man with a history of myelodysplastic syndrome, type 2 diabetes and arterial hypertension was admitted to the Respiratory Care unit of our hospital with a diagnosis of SARS-CoV-2 pneumonia. The patient developed acute respiratory distress. On day 5 he underwent non-invasive ventilation with Continuous Positive Airway Pressure (CPAP) and a treatment with intravenous corticosteroids, initially with dexamethasone 6 mg every 24 h (two-day course) and then with methylprednisolone 40 mg every 12 h (25-days course with subsequent decalage). Given the sustained fever, the altered biochemistry indexes (C reactive protein 31.65 mg/dL) and his myelosuppression (neutrophil count 1270 /mmc), the patient was treated with multiple empiric antibiotic lines (piperacillin/tazobactam, cefepime and ceftobiprole). Finally, on day 27, he was treated with daptomycin and ceftaroline for *Enterococcus faecium* bloodstream infection, with a subsequent shift to vancomycin and ceftaroline for the lack of clinical response. On day 30, a transthoracic echocardiography was performed, excluding valvular vegetations compatible with endocarditis, whereas the thoracic high-resolution computed tomography (HRCT) performed on day 38 showed pseudo-nodular cavitary consolidations in the right apical zone (Figure 1A). The CT scan of facial bones only showed an inflammatory thickness of the left maxillary sinus. In the suspect of COVID-19 associated pulmonary aspergillosis (CAPA), voriconazole iv (300 mg every 12 h) was started. *Aspergillus fumigatus* susceptible to azoles (the MIC of voriconazole was 0.094 mg/L) was isolated on samples of broncho-alveolar lavage performed on day 39, and antibiotic therapy with vancomycin and ceftaroline was stopped. Due to the lack of clinical improvement (the need for oxygen therapy with high-flow nasal cannula and the persistent fever), on day 53, therapy was then maximized, adding anidulafungin (200 mg the first day, then 100 mg/day) to voriconazole and introducing wide range empiric antibiotic therapy with linezolid (600 mg every 12 h) and piperacillin/tazobactam (3.50 g every 8 h).

The patient was then transferred to the Infectious Diseases (ID) department because of the worsening pulmonary radiological pattern and the persistence of fever. On day 55, the patient underwent a fundus oculi exam, which resulted negative for local infections. Blood cultures were performed at different times during his ID ward stay, and they were all negative. Blood cultures for mycobacteria and blood samples for galactomannan, β-D-glucan, HSV-1 DNA, HSV-2 DNA, CMV-DNA, EBV-DNA and BKV- DNA, serologies for *Toxoplasma gondii* and *Strongyloides stercoralis* and bacteria associated with atypical pneumonia were also collected and yielded a negative result. On day 58, the patient underwent a bone marrow biopsy and bone marrow aspiration, tested negative for CMV, EBV and *Parvovirus* B-19 DNA, *Leishmania* spp. PCR and bacterial and mycobacterial cultures. The same day, gas exchange deteriorated and respiratory distress increased. Non-invasive ventilation with CPAP was then restarted. A thorax CT scan showed a ground-glass and crazy-paving pattern (Figure 1C). Therapy with methylprednisolone and trimethoprim-sulfamethoxazole (STX) was then started due to the clinical and radiological suspicion of *Pneumocystis jirovecii* pneumonia, despite the ongoing prophylaxis with atovaquone that was started because of myelodysplastic syndrome and corticosteroids therapy. Moreover, linezolid was stopped and meropenem was substituted with piperacillin/tazobactam. On day 59, a bronchial aspiration (BAS) was performed, and a Mucorales (identified after some weeks as a *Mucor circinelloides* resistant to voriconazole and susceptible to amphotericin B with a MIC of 0.094 mg/L) was identified. On day 61, the antifungal therapy was changed and the patient was administered with amphotericin B (7 mg/kg/day) and caspofungin (50 mg/day). The search for *Pneumocystis jirovecii* DNA on BAS material was negative; therefore, STX was stopped and steroidal therapy was reduced. The patient further deteriorated, with worsening hypoxaemia and dyspnoea despite maximized CPAP support, and he finally died on day 64.

## 4. Discussion

Different incidence rates of CAPA have been described in the literature, from 0.7–7.7% among COVID patients to higher percentages among ICU patients and people with severe COVID-19 pneumonia [12,18]. The more frequent baseline conditions associated with CAPA are diabetes mellitus, corticosteroid administration, chronic cardiovascular diseases, renal failure and obesity [12,19]. As highlighted by our case, the clinical management of fragile patients with COVID-19 represents a difficult challenge. The coexistence of an impaired immune system alongside comorbidities often hampers differential diagnosis and the correct clinical and therapeutic approach. In this setting, SARS-CoV-2 infection can be the initial trigger of respiratory failure, and it favors infective complications. This can be related to a direct effect, consequently to airway epithelium damage and gut microbiota composition alteration, which favors fungal superinfection. Or, it can happen indirectly due to prolonged hospitalization, ventilatory support and the long courses of corticosteroid therapy required in the management of COVID-19 patients [20]. For these reasons, it is crucial to be aware of such risks, monitoring these patients with even further precaution and eventually performing regular screening for opportunistic infections (e.g., fungal biomarkers such as β-D-glucan and galactomannan). Furthermore, in the case of unresponsiveness to standard care for COVID-19, the clinical workup should be extended to extensive cultural enquiries (e.g., bronchoalveolar lavage) in addition to thoracic and paranasal sinuses imaging, aiming at the early recognition of microbiological and radiological signs of fungal disease. In conclusion, since SARS-CoV-2 continues to be a worldwide concern, other data are needed to better assess the incidence and the management of fungal superinfections in COVID-19. Prompt diagnosis is paramount to start appropriate antifungal therapy and, as emphasized by our experience, shape the clinical course of fragile COVID-19 patients.

## Figures and Tables

**Figure 1 jof-07-00708-f001:**
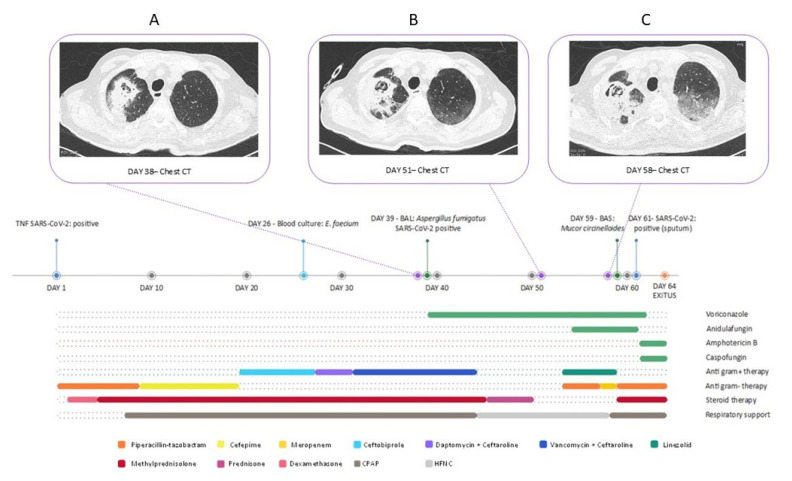
Timeline of the case, highlighting the therapies administered, the respiratory support provided and the evolution of the pulmonary lesion (**A** chest CT scan at day 38 since diagnosis, **B** chest CT scan at day 51 since diagnosis, **C** chest CT scan at day 58 since diagnosis).

## Data Availability

Data will be provided on reasonable request.

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
