# Peer review of "Mucormycosis in CAPA, a Possible Fungal Super-Infection"

_jof, 2021, doi:10.3390/jof7090708_

Round 1

Reviewer 1 Report

The submitted manuscript is an interesting case report about a dual fungal infection in a COVID-19 patient.

It is vell organized, but certain issues must be addressed.

At first, in the title the authors describe the case as a co-infection, but in the text it becames a super-infection: please better define this condition.

In addition, the authors report that the patient had a "...Enterococcus faecium bloodstream infection..." (line 77). Can this infection facilitate the fungal superinfection, and the worsening of the patient status? Please discuss this in the light of literature data about COVID-19 and human microbiota (i.e., 10.2174/1871530321666210127141945, doi: 10.20463/pan.2021.0008)

Line 42: mucormycosis is so not rare in COVID-19 patients.

Line 61: add the acronym MALDI-TOF; the word "life" must be corrected as "flight"

Lines 64-65: please discuss the statement in the light of literature data (for example, https://doi.org/10.3390/jof6040343; doi: 10.3390/jof7070552)

Please add a Conclusion section after the Discussion

At last, a revision for typos and English language and grammar is needed.

Reviewer 2 Report

Many CAPA cases have been described so far, so as several CAM, however the combination of both is not that frequent, therefore, I think this case report is quite interesting.

Nevertheless, there are certain aspects that need to be solved, in my opinion, before its acceptance for publication.

INTRODUCTION

- "Mucorales" is a genus order and not a genus, therefore "Mucorales spp." is innacurate, only "Mucorales" is enough.

- "Aspergillus spp." is accurate, although the "spp." does not need to be in italics.

- In line 31, there is a mention to bacterial and fungal co-infections in COVID-19 patient, but there are no references. I would suggest to include some, specially those with the same bacteria/fungi as your patient.

- Additionally, since some denominators have been published already regarding CAPA, at least, I would suggest to mention them, so the reader can have a better idea on the incidence, mortality, etc. Good ideas are Bartoletti et al. (an Italian site) or Salmanton-García et al. (global research with general, ICU and MV incidence).

- Lines 41-42: The references are regarding only CAM, I would suggest to include also CAPA references, since there is a mention to it.

MATERIALS AND METHODS

- Please provide the city and the country of each of the manufacturers.

RESULTS

- In order to facilitate the comprehension of the chronological order, I would suggest to implement in the text this kind of structure "On day x, ...", "on day y...", "on day z...". This makes that the people can understand much better the time between the different events described in your text.

- Line 83: Could you provide the formulation of the voriconazole (po/iv), so as the dosages?

- In the text, there is mention to the CT, I would suggest to linked them to the figure; something like this "...CT (Figure 1A) [...] CT (Figure 1B)"

DISCUSSION

- There is no reference in your discussion. I suggest that you compare your case with what it has been described in the literature so far. Additionally, I think that in its current version is more a summary/conclusion paragraph, rather than a discussion itself.

FIGURE

- I would suggest to include letters in the CTs, so you can mention within the text.

- Please include the dosage, so as the formulation for the drugs (especially the antifungals) both in the figure and in the text

- Please, improve the image quality. Additionally, I would suggest to make it a bit bigger, so all the details can be better perceived.

Reviewer 3 Report

General comments

This short report describes the case of a 72-year old male patient, COVID positive, that develops pulmonary aspergillosis and mucormycosis. Despite the administration of different antifungal combination therapies, and despite the fact that the fungal strains isolated are susceptible to (some) antifungals, the treatments do not work and the patient passes away 64 days after hospitalization. The patient had several other multiple underlying conditions that made this case difficult to treat.

The authors emphasize the importance of monitoring closely risk patients, such as immunocompromised, since they are more susceptible to secondary opportunistic infections. However, my main concern with this manuscript is the lack of discussion and references of other cases of fungal-COVID co-infections. Recently, a case of co-infection with Rhizopus and Aspergillus in a COVID patient was published and there is no mention of it here. Also, there are multiple cases of coronavirus disease and aspergillosis or mucormycosis, but they are not commented here. Both introduction and discussion could be more complete and detailed.

Specific comments

Line 29 - …mellitus OR cancer.

Line 44 - …agents ARE more…

Lines 63-64: Etest was employed to determine antifungal susceptibility to what antifungals? Please, specify.

Lines 64-65: there are no clinical breakpoints, but there are Epidemiological cutoff values for several antifungals and different species of Mucorales and Aspergillus.

Line 83: replace “sensible” by “susceptible”

Line 108: was the M. circinelloides isolate susceptible or resistant to voriconazole?

Line 108: replace “sensible” by “susceptible”

Round 2

Reviewer 1 Report

The authors were very clear in addressing queries. I think the paper is now eligible to be published.